# The Correlation between the Number of Asphalt Mixing Plants and the Production of Asphalt Mixtures in European Countries and the Baltic States

Henrikas Sivilevičius [1] , Viktor Skrickij [1] and Paulius Skačkauskas [2],*

1. Transport and Logistics Competence Centre, Faculty of Transport Engineering, Vilnius Gediminas Technical University, 10223 Vilnius, Lithuania; henrikas.sivilevicius@vilniustech.lt (H.S.); viktor.skrickij@vilniustech.lt (V.S.)
2. Department of Mobile Machinery and Railway Transport, Faculty of Transport Engineering, Vilnius Gediminas Technical University, 10105 Vilnius, Lithuania
* Correspondence: paulius.skackauskas@vilniustech.lt

**Abstract:** The transport infrastructure's pavement is made of asphalt layers, placed and compacted. The asphalt mixture is produced in an asphalt mixing plant (AMP) using expensive technological equipment which, when it becomes obsolete and worn out, is replaced with new equipment. One of the main problems related to the replacement process is that when it comes to purchasing new AMPs, the decision making involved is, in most cases, highly intuitive due to a lack of clearly defined criteria. In order to remedy this situation, this work presents an analysis of the correlation between the number of AMPs and the production of asphalt mixtures. Firstly, a correlation analysis was performed based on the European Asphalt Pavement Association (EAPA) data. Secondly, the current situation with the AMPs in European countries was analysed. Furthermore, a case study was performed and a system of nine criteria was created to identify why/when road construction companies operating in the Baltic States buy new AMPs. The weights of the criteria have been established by applying the Analytic Hierarchy Process (AHP) method. It was found that the most important criterion during the decision-making process for road construction companies is increased requirements for improving the quality of the asphalt mixture produced (criteria weight 25.0%). With a weight of 20.6%, the second vital criterion is the possibility of receiving support from European funds. The third criterion is the expectation of having a sizeable asphalt-paving site (weight 20.4%). The other six criteria are also significant, their weights varying between 1.6% and 13.5%. The industrial companies can use the obtained results for designing, producing and selling AMPs and adjusting strategic business plans.

**Keywords:** asphalt mixing plant; asphalt mixture; correlation; production; causes for purchasing; AHP method

## 1. Introduction

Roads, streets and airfields are constructed using an asphalt mixture produced in an AMP. An AMP consists of a set of related equipment that performs the operations of the technological process of asphalt mixture production. Preparatory operations include procedures for handling virgin mineral materials, metered supply, drying and heating. The mixture of dried and heated aggregates in a batch AMP is additionally screened into separate hot fractions, which are finally dosed based on a mass summation principle. During the technological process in the drying drum and screens, the smallest particles separate from the aggregates, which are then collected in the baghouse as reclaimed dust (RD). The RD, together with an imported filler (IF), is dosed on a separate scale. In the end, the bitumen heated to the working temperature is dosed. In recent years, the use of reclaimed asphalt pavement (RAP) for the production of hot mix asphalt (HMA) mixture has increased significantly. The RAP must be processed, dried and heated by applying conduction or convection methods. After adequately completing all these preparatory

technological operations, the hot aggregate fractions, IF with RD, hot bitumen and RAP are dosed into the mixer using separate scales. The primary operations of forming the structure of the asphalt mixture are carried out in the AMP mixer. The bitumen is transformed from the volumetric state into films covering all the particles. The different-sized mineral particles are evenly distributed in the asphalt mixture. The asphalt mixture produced in the rotating mixer is handled into a truck or a storage silo. All the technological operations during the asphalt mixture production process and their parameters will have an impact on the quality of the asphalt mixture as well as the structure and properties of the asphalt pavement. During the production of the asphalt mixture in the AMPs, the production costs and environmental pollution need to be minimised. However, at the same time, the technological universality and reliability of the AMPs should be increased.

Each country has a certain number of AMPs capable of producing a specified amount of asphalt mixture. This amount depends on the available investments in road construction, the intensity of the road network development, the AMP mixing capacity, the climate and weather conditions. Due to unavoidable wear of AMPs, stricter requirements for the quality of asphalt mixtures and ambient air pollution, the old equipment in the AMPs is replaced with new equipment or new AMPs are installed on new sites. As the manufacturers of AMPs can produce any device that meets the needs of a potential buyer, the decision to purchase a new AMP must be made very responsibly. However, one of the main problems is that AMPs are usually bought very intuitively, without a clearly defined fleet renewal strategy. Thus, this paper describes the adoption of a scientific approach while seeking to define the most important criteria for the AMPs' fleet renewal strategy. The obtained results can be used by the stakeholders, such as business companies and public authorities, to fill the gap addressing the AMPs' fleet development issues.

Respectively, the research framework diagram of this study is shown in Figure 1. From the provided diagram it can be seen that the research was performed in three main steps. With the first step, a model for the theoretical substantiation of the required number of AMPs in the country was developed. This model considers the number of AMPs in the country and the expenses related to the AMPs and road transport users. Furthermore, detailed analyses based on the developed model and the statistical data from the EAPA about asphalt mixtures and the number of AMPs in different European countries, were performed for European countries and the Baltic States. Step two is specifically related to the situation in European countries and presents the analysis which consists of four parts: (1) statistical analysis of the amount of asphalt mixture and the number of asphalt mixing plants; (2) analysis of the correlation between the number of AMPs and the amount of asphalt mixture produced in those AMPs; (3) categorizing of the countries according to the produced amount of asphalt mixture and; (4) determination of the average amount of asphalt mixture annually produced in a single AMP for different countries. The results obtained during the second step were the basis for the further analysis of the Baltic States. Similarly, the third step—the case study of the Baltic States—was also divided into four parts: (1) analysis of the historical data; (2) description of the possible AMPs' fleet renewal cases; (3) discussions with the authorised persons from the companies operating AMPs in the Baltic States, and the proposition of a novel nine criteria system which determines the decision-making process to update the AMPs' fleet; (4) determination of the significance of each proposed criteria while using the AHP method. The historical data was analysed in order to present the case of the Baltic States in as much detail as possible. The most important part is the third step—the proposition of the nine criteria system and the determination of the significance of those criteria. These nine criteria and their significance, which were defined on the basis of a consideration of the Baltic States, is the main result of this study and can serve as a set of guidelines for the AMPs' fleet renewal strategy planning and decision-making process.

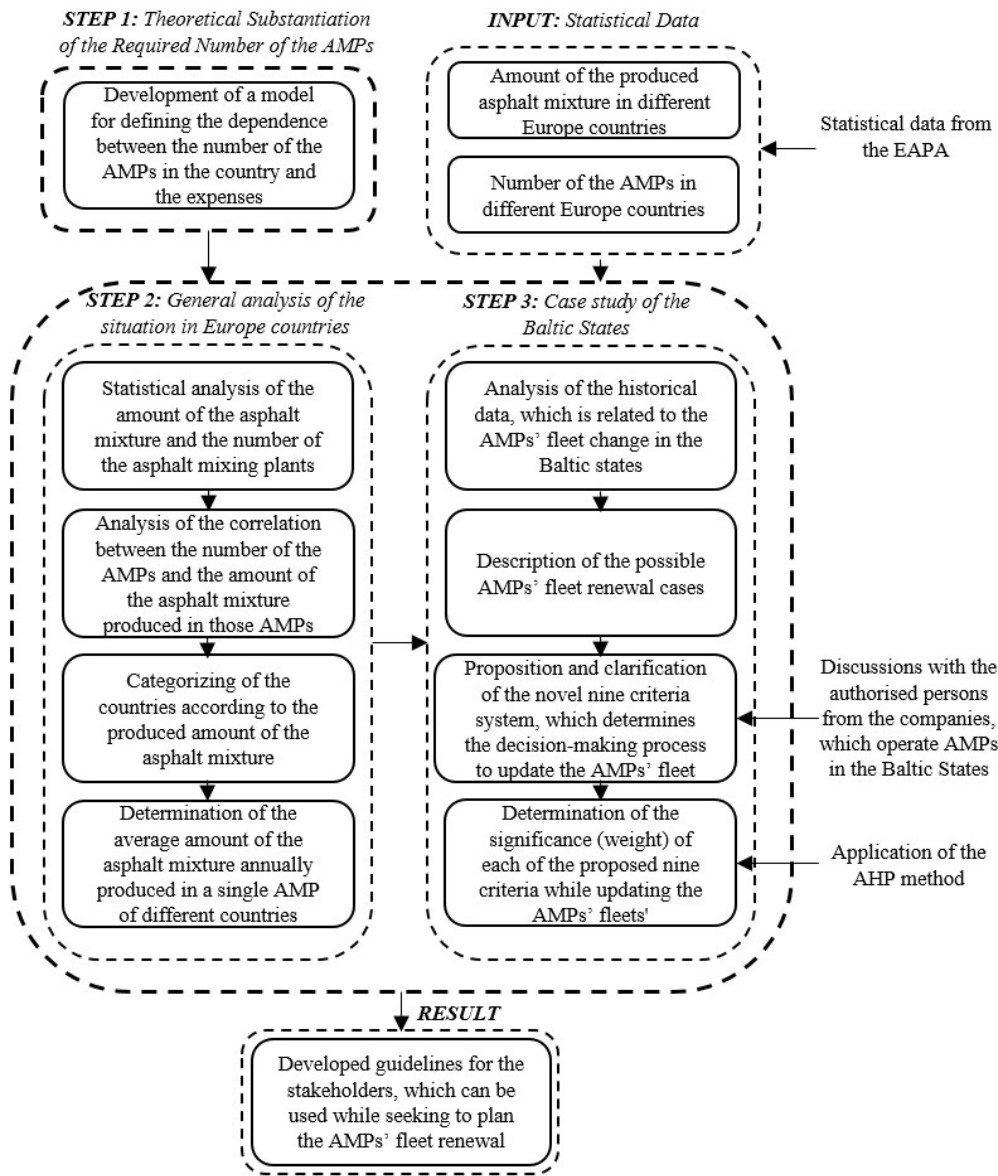

**Figure 1.** The research framework diagram of the study.

While taking into consideration the described research framework, this paper is organised as follows: in Section 2, a literature review is performed, all the stages of the technological process of the asphalt mixture production are examined, and the AMPs' energy consumption and environmental pollution problems are reviewed. In Section 3, the current situation with the AMPs in European countries is analysed, the correlation between the amount of asphalt mixture produced and the number of AMPs in different countries is found. As a result, an AMP capacity utilisation rate is defined. In Section 4, a case study is performed and the analysis of the current situation of the AMPs' fleet in the Baltic States is provided. The criteria system for the decision-making, whether to purchase an AMP or not, is proposed and described.

## 2. Literature Review

The performance requirements for road-building machinery, including asphalt-mixing machines (AMMs), are continuously increasing. Road quality depends on the operation of the metering, batching and combustion systems of the AMMs [1]. According to standard [2], while considering the principle of the production of the asphalt mixture, AMPs fall into the

following categories: batch, continuous and drum mix plants. The most common are the drum mix and batch plants [3].

In most countries, the volumes of asphalt mixture produced for road pavements tend to increase. Currently, the majority of roads in the United States (US) have asphalt surfaces. The exact number of HMA plants in the US is unknown; according to the literature [4], there are about 3600. In [5], the authors state that the number of the hot mix asphalt (HMA) plants is equal to 4500; in Europe, there are about 4000 HMA plants. According to the National Asphalt Pavement Association of the US, the investment in the transportation infrastructure accounts for 7% of the Gross Domestic Product. These investments increase every year because of pavement durability, safety and efficiency, requiring modified materials due to the increased number of vehicles and their loads [6].

Asphalt production is a highly energy-consuming process [7]. Asphalt concrete includes mineral components and a bituminous binder that makes the hydrocarbon material. Before mixing the aggregates, they need to be dried and heated. These procedures are performed in the rotary drum and have never been studied from a physical perspective [8]. Bitumen-aggregate mixtures for pavements have been produced for many years using conventional batch mixing equipment [9]. Hobbs, in [10], used a combination of the computational fluid dynamics and the discrete element method to simulate the thermodynamic processes occurring in the drum dryer of an AMP. Therefore, the minimisation of the energy consumption for manufacturing HMA and the stricter requirements for the asphalt plant emissions have become essential issues [11]. Ang et al. [12] combined a study on the process of energy usage in two AMPs. It was found that a large part of the energy was used for drying and heating the aggregate. The statistical analysis of the historical data was provided; the analysis of the data on the energy consumption showed a high correlation between specific energy consumption and the rainfall level. Androjic et al. [13] suggested that a possible way to achieve energy efficiency and, therefore, sustainability is to preheat the mineral mixture before it enters the production process in the asphalt mixing plant, thus resulting in lower energy consumption per ton of asphalt. Grabowski et al. [14] identified factors that affect the energy consumption during the HMA production by focusing on the amount of fuel needed for drying and heating the aggregate while considering their moisture as well. It was found that energy consumption can be reduced by approximately 15% by decreasing the moisture content only by around 1%. In [8], the authors analysed the drying and heating processes when the aggregate and hot gases were involved in the co-current flow. This step of the process requires a large amount of the overall energy needed for HMA manufacturing. The analysis of the energy and exergy in a rotary dryer employed in a HMA plant for heating and drying the aggregate in the mixture is presented in [4].

Many emissions are released into the environment during the production of road pavement materials [15]. In [7], the authors presented the idea of the open life cycle assessment framework, which allows the quantification of the potential environmental impacts due to the construction, maintenance and disposal of roads. It was found that the asphalt production stage and the transporting process were the two most energy-consuming and most polluting stages. Gschösser et al. [16] achieved similar results; the authors found that pavement construction and deconstruction have less impact on the pollution than the material production, transporting and recycling processes. Furthermore, Wang et al. [17] noted that the volatile organic compounds generated during asphalt pavement production have a negative impact on the atmospheric ecosystem and that this problem has raised increasing concerns globally in recent decades. Itoya et al. [18] also revealed that asphalt production, compared to other processes, led to the most significant carbon emissions. The greenhouse gas (GHG) emissions are an essential primary criterion for environmental evaluation [19]. From the case studies provided in [20], asphalt production was identified as the process generating the most GHG emissions. Kim et al. [21] presented a comparative analysis of GHG generation while employing various types of equipment used in different construction activities. In the study by Kim et al. [22], a framework was developed for

estimating GHG emissions for land-use changes. Due to the necessity of heating and mixing the aggregate and binder [15], a HMA plant uses more than half of the energy and generates a similar amount of global warming emissions in the production of materials. The asphalt industry is continuously attempting to reduce its emissions due to the problem of global warming. The emissions can be reduced during mixing by decreasing the temperatures and compaction processes without affecting the quality of the mixture. This is realised by implementing various available industrial technologies [23], for example, by using warm-mix asphalt (WMA) technology at 120 °C or lower temperatures; for HMA, 160 °C and higher temperatures can be used. Li et al. [24] performed comprehensive tests and stated that, in view of the characteristics of the mixing plant and the temperature drop during the construction procedure, the following construction temperatures are recommended: an aggregate heating temperature of 155 °C, a mixing temperature of 150–160 °C, a discharging temperature of 150–160 °C, a laydown temperature of 145–155 °C, an initial compaction temperature of 135–145 °C, a secondary compaction temperature of 130–140 °C and a final compaction temperature higher than 90 °C. Cucalon et al. [25] also noted that reduced temperatures provide a greener technology as emissions are reduced at the plant and the construction site. The mixing temperature for binders is usually chosen by a pavement engineer based on the specific "viscosity" and gradation of the HMA. The pollutant emissions are evaluated in a lab by applying a comprehensive experimental design [26]. As the HMA manufacturers are required to meet the airborne emission thresholds, it is essential to define ways to measure those emissions.

In [27], the authors investigated the influence of the equipment operating parameters on the airborne emissions, focusing on gaseous organic compounds. Rubio et al. [28] presented a study in which an analysis of the environmental benefits derived from cleaner production technology for manufacturing asphalt mixtures is provided. The air pollutant emissions from the construction operations can be evaluated by estimating, benchmarking and monitoring at the pre-construction and construction stages [29]. The particulate matter emissions filtered out in a pulse-jet filter unit were analysed in [30]. The separation efficiency of the cyclone separator was investigated in [31]; the authors used an inlet flow with particle concentration within a range from 5 to 2000 mg/m$^3$ for the research. It was found that the efficiency of the device increases as the inlet flow gets higher. The emission problem, while using numerical modelling, was investigated in [32]. A multichannel cyclone is a new-generation cleaning device that purifies gas flow and removes dispersion solid particles of 2 μm and those larger in diameter from it. The hot fractions of the aggregate screened in a batch-type AMP segregate in bin compartments. The larger the number of the hot aggregates being screened, the narrower the less segregated fractions. Currently, the screener of an AMP screens six hot fractions. The dynamics of the segregation processes of the minerals' hot fractions in the hot bin compartments depend on the compartment size (volume) and form, the degree to which the volume is filled with the screened material, the fluctuations in its content and compartment frequency. The more each hot fraction is contaminated with grains smaller and larger than those of the established nominal (required) size, the more variations can be observed in its gradation [33]. The hot-bin storage needs to be of sufficient capacity. There should be a specified number of bins to ensure the separation and storage of the aggregate fractions. Additional facilities are required for obtaining aggregate samples [2]. To reduce the segregation of the mineral materials in the hot bin compartment, installing a baffle is recommended. The baffle causes dust to slide to the centre of the bin, where it is uniformly mixed with coarse materials [34].

The function of the bitumen batching system is to batch the substance heated up to a required temperature and spill it into the mixer. The bitumen oxidation process is dependent on the thickness of the bitumen film coating the aggregate particles, the temperature and the short-term ageing time. When the bitumen is flowing into the mixer due to gravitation only, the oxidation process is more intensive than it is with the technology where the bitumen is batched by a high-pressure pump [35]. Respectively, Bražiūnas [36] noticed that, while increasing the mixing time, due to accelerating oxidation of the bitumen,

the most important normative physical and mechanical parameters of the asphalt mixture can be improved.

The asphalt industry must remain economically competitive; the use of available materials and transportation costs must be minimised. The usage of locally available materials reduces these costs, and the reuse of reclaimed asphalt pavement (RAP) decreases the quantity of the new aggregate [37]. Reusing old asphalt concrete in a plant to produce new asphalt is cost-effective [38]. In a plant recycling the HMA, the calculated quantity of the new aggregate and asphalt binder is added to the RAP and the new asphalt mixture is produced [39]. Hot-mix recycling in a plant has two main benefits. The first benefit is the price of the final product. The second one is that such technology generates fewer emissions during the production process. The economic effect is achieved by minimising the new aggregate and bitumen needed for the production [40]. Ding et al. [41] evaluated the recycling efficiency of plant-asphalt mixtures containing RAP and concluded that, during the production process, only a part of the RAP binder is being mobilized, i.e., the RAP binder mobilization rate is lower than 100%. During the research, it was also noticed that there are many other important factors, like temperature, that affect the mobilization rate. However, the mechanisms of the thermal production conditions and their influences on the performance of RAP mixtures are not well understood [42]. For example, Ma et al. [43] found out that increasing the heating temperature of RAP is not always an effective method to improve the compactability of recycled HMA. The recycled HMA mixtures containing 20% RAP are easier to compact with RAP heated to 120 °C than the same mixtures with RAP heated to 140 °C. Sivilevičius and Vislavičius [44] proposed algorithms that were developed by using mathematical programming methods, which helped to minimise the bitumen content and, as a result, reduced the final product price. Extensive efforts have been made to use old pavement for new asphalt production in plants [45]. A control system is needed to meet the quality requirements [46]. In [47], the authors analysed asphalt recycling technologies and provided a comparative analysis. Mogawer et al. [48] investigated two types of plants (batch and drum) and provided an analysis of the mixture performance and the degree of blending between the short-term aged and virgin binders.

During the mixing process of the RAP virgin aggregate and virgin binder, partial blending occurs. The RAP is usually limited as the degree of blending between the virgin materials and the RAP is not known [49]. In [50], the authors found that a complete blending may be reached when the mixing time is increased. However, a longer mixing time could result in a final product with a higher stiffness modulus and better homogeneity. The paper by Eddhahak et al. [51] presented a generalised self-consistent scheme-based micromechanical model for investigating the mechanical behaviour of the high-rate recycled HMA mixture. With a growing amount of milled RAP in developed countries, techniques and technologies that allow the highest possible percentage (up to 100%) of the RAP in the recycled asphalt mixture are developed [52]. Design considerations of high RAP-content asphalt produced at reduced temperatures were described by Abed et al. [53]. However, the results of the study suggested that the full blending hypothesis is unpractical for the sake of asphalt plant productivity. The maximum content of the RAP allowed in a recycled HMA depends on its homogeneity [54]. In [55], the authors presented an algorithm for predicting the composition of the mineral constituent of recycled HMA, taking into account the gradation of the virgin mineral materials and RAP. In [56], research was undertaken to evaluate the production and construction possibilities of hot-mix asphalt concrete (HMAC) containing large quantities of RAP. In general, a higher variability was found in the HMAC produced when RAP is excluded than in the typical HMAC product. The commonly used means of incorporating RAP into asphalt production is by initial superheating of the virgin aggregate. After that, RAP is dried and heated by conduction. Such an operation is needed when the RAP content is more than 40%. The use of a double dryer or a parallel drum for heating the RAP and virgin aggregate allows the process to be carried out without a direct flame [57]. The ability to thoroughly heat the RAP particles is based on the experience of the relevant personnel [58]. For producing the asphalt mixture in an AMP, different addi-

tives and rejuvenators [59] which improve the blend anti-stripping properties, are used; to receive, store (keep), supply and dose them, additional AMP equipment is required. Hesami and Mehdizadeh [60] studied the effects of the amine-based liquid anti-stripping additives to HMA production. The results of the study indicated that the effectiveness of this type of additive significantly decreases after being heated long-term for HMA production. Thus, the researchers suggested that, during the HMA production process, the percentages of liquid anti-stripping additives used should be carefully examined with regard to production conditions. While taking into consideration the use of quarry dusts and industrial by-products, Valentin et al. [61] concluded that quarry dusts, blast furnace slags or selected construction and demolition wastes can be incorporated into HMA to act as fillers. During asphalt production, a rejuvenator can be added to restore the RAP binder. Ten possible locations for adding the rejuvenator in the AMP were investigated by [62]; the authors took into account such parameters as pavement performance, plant operation and environmental protection. In [63], the authors evaluated the effects of the addition of the rejuvenator to the cold RAP before the RAP heating drum. In this case, ten potential locations for adding the rejuvenator were investigated as well [63].

Many AMP plants store the mixture in silos before transportation. The time the material is stored in the silo depends on several factors and may vary. The short-term ageing of the binder may occur due to high temperatures [64]. As a result, the pavement service life may be shorter [65]. A pavement oxidation model was developed and investigated in [66] by using simulation tools.

The quality of the HMA mixture depends on the quality of the input aggregate, its gradation, moisture level, the AMP itself and the production process control [67,68]. The operations of the technological process of manufacturing the asphalt mixture are under the control of its parameters. The control system for the batch asphalt mixing equipment, measurement and gradation was described in [1]; the authors introduced a solution to the factors affecting the quality of the asphalt mixture.

In [68], the authors proposed a multi-attribute model for the quality assessment of an AMP. In [69], the authors developed a quality system for an AMP based on original statistical methods. A review of the history of the coating plant in the UK is followed by a look at the recent developments which have improved control and environmental protection [70]. Data on the power inputs, costs and production of asphalt concrete mixtures within a year were cited in [71]. It showed a reduction in the temperature at which a mixture is prepared, which was the most effective means of decreasing power inputs. Thus, the most outstanding economy can be obtained with lowered air temperatures and a low mixing installation load. Lupanov and Gladyshev [72] provided data on pollutant emissions resulting from the production of asphalt concrete mixtures and the fee for the emissions from mixing and resetting oil-fired gas-fired boilers.

The literature review shows that the production of asphalt mixtures is a very complicated technological process, which is directly related to an AMP's energy consumption and environmental pollution problems. Respectively, the technological process of the asphalt mixture production has a significant influence on an AMP's ability to produce the specific required amount of high-quality asphalt mixture while minimising environmental pollution and while still seeking to achieve financial benefit. As both the insufficiency and the excessive amount of the asphalt mixture produced are undesirable possible results of the production process, the results of this work will allow stakeholders to plan their AMPs' fleet renewal strategies more precisely, while taking into consideration the requirements of the asphalt mixtures to be produced.

In this paper, the amount of asphalt mixtures produced and the number of AMPs across Europe is analysed. A case study for the Baltic States is also presented. The analysis allows the calculation of the utilisation level of an AMP's capacity and an evaluation of the significance of the criteria used for making decisions about the purchase of new AMPs. The weights of the criteria are found using the AHP method.

## 3. The Correlation between the Amount of Asphalt Mixture Produced and the Number of Asphalt Mixing Plants

### 3.1. Theoretical Substantiation of the Required Number of Asphalt Mixing Plants

Mixing capacity (MC) is one of the most critical parameters for an AMP. The maximum hourly MC of an AMP depends on the structure and dimensions of individual units that make up the AMP, these units being selected by the engineer and manufacturer according to the preferences of the purchaser. The MC is usually listed in the technical specifications of the equipment concerning specific conditions. The MC of an AMP shows the ability to provide the companies with an appropriate amount of asphalt mixture, which usually depends on the actual MC of the asphalt paver when the mixture is laid evenly. AMPs with a MC from 150 to 400 t/h are most common. A low temperature of ambient air and materials, high humidity and the reduced content of the mineral materials (aggregate) supplied to the drying drum to increase the efficiency of screening them into the hot fractions reduce the actual MC of an AMP. To improve the homogeneity of the asphalt mixture, the time period for mixing all the AMP materials is extended; however, this decreases the MC. A decrease in the MC of an AMP is observed while increasing the RAP dose through a conductive heat transfer from the superheated aggregate when the hotter aggregate comes into contact with the cooler RAP granules.

The number of AMPs nationwide correlates with the required and actual amount of asphalt mixture produced by the country in question. The larger the area and population of the country, the better developed the road transport infrastructure, the higher the geometric parameters for roads and city streets, the more enhanced the quality of their asphalt pavement and the more asphalt mixture needs to be produced. Thus, growth in the required amount of asphalt mixture increases the number of AMPs across the country. The developed cost model (Figure 2) shows that as the number of AMPs $\overline{N}_c$ grows, the expenses of purchasing and maintaining those AMPs increase as well. It also results in a decrease in the costs for the road users. Many AMPs can produce a large amount of the asphalt mixture used for improving the quality of the road transport infrastructure. A proper condition of the asphalt pavement and the type of pavement introduced on gravel roads reduce the transportation costs, environmental air pollution and road accidents.

In Figure 2, the upper curve (3) shows the public expenditure (*c*) obtained, summing up (*a*) the costs necessary for purchasing and maintaining the AMP (curve 1) and (*b*) the costs of road transport users (curve 2). This curve (3) has the lowest cost value when the country has an optimal number of AMPs $\overline{N}_{c.opt}$. In this case, if there is a minimal number of nationally owned AMPs ($\overline{N}_{c.min}$), the road transport infrastructure is likely to be of low quality, which leads to high public expenditure. The national surplus of AMPs ($\overline{N}_{c.max}$) also increases the public expenditure. This is because a considerable investment is usually made for purchasing the AMPs and the acquired equipment needs to be properly supervised, even though the asphalt mixture is not produced and added value is not created.

The road network usually meets the required parameters in economically developed countries. A rise in the level of automation, a drop in the number of traffic accidents on urban streets and roads and the wear of the asphalt pavement due to traffic loads and weather conditions lead to a certain amount of the asphalt mixture being produced each year. The produced HMA and WMA mixture is not stored and, therefore, in a sufficiently short time (within a few hours), is laid in the layer of the pavement. The total quantity of the asphalt mixture produced in all AMPs of the country is equivalent to the quantity required. It is determined by the funds annually allocated for the Road Maintenance and Development Program. The developing economies make smaller investments in road maintenance and development than is required to preserve appropriate road pavement conditions and build new roads. An insufficient amount of the asphalt mixture manufactured can also be determined by the lack of AMPs operating in the country and their low MC when the outdated technological equipment is not replaced with modern AMPs for economic reasons.

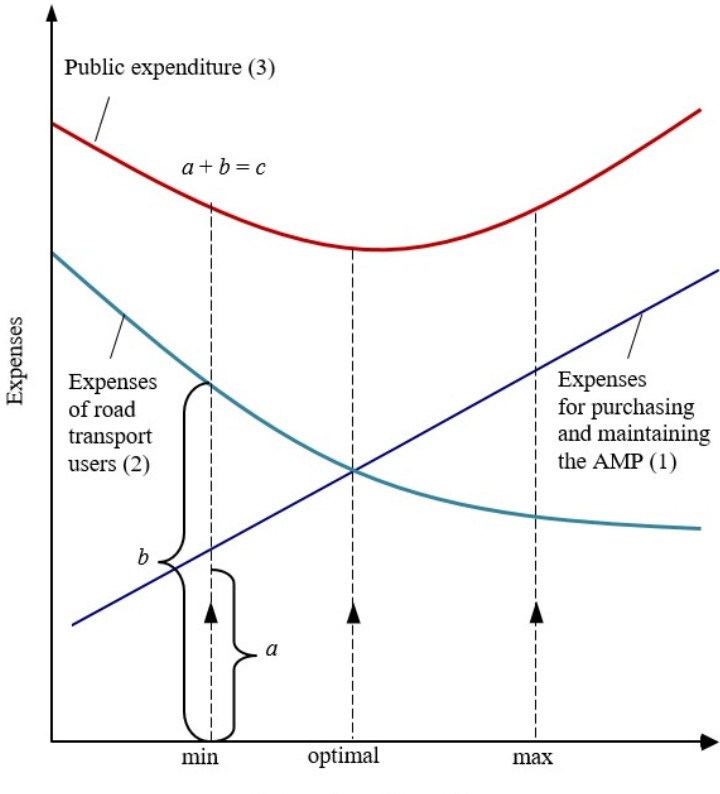

**Figure 2.** The model of the dependence between the number of AMPs in a country and expenses.

The MC of many AMPs allows for the manufacture of much more asphalt mixture than is produced and consumed each year in most countries. Therefore, some AMPs are inefficiently used because of the extended downtime that is not always due to adverse weather conditions. In certain months of the year, an uneven demand for the asphalt mixture makes it necessary to increase the number of AMPs so that their production capacities could be sufficient to meet demand in the case of a sudden increase in the scope of asphalt paving (work volume). The colder the climate of a country, the greater the difference between the amounts of the asphalt mixture produced over the months and during different seasons.

*3.2. Statistical Analysis of the Amount of Asphalt Mixture Produced and the number of Asphalt Mixing Plants*

According to the overview *Asphalt in Figures 2005 to 2016*, publicly provided by the EAPA every year [73], the correlation between the amount of asphalt mixture produced in different European countries and the number of AMPs operating across Europe has been determined. The collected data refer to asphalt production, application, binders and the number of production sites.

Initially, the total production of HMA and WMA mixtures in metric tonnes for each of the 30 European countries included for one calendar year for the period from 2005 to 2016 has been calculated:

$$\overline{P}_c = \frac{\sum_{y=1}^{Y} P_{yc}}{Y},$$

(1)

where $P_{yc}$ represents the total annual production of the HMA and WMA asphalt mixture in the *c*-th country (millions of tonnes); $c = 1, 2, \ldots C$, for the number of countries included in the study, with $C = 30$; and $Y$ is the number of years ($y = 1, 2, \ldots Y$, with $Y = 12$).

In addition, the average number of production sites, i.e., the number of AMPs operating in every *c*-th country, has been calculated:

$$\overline{N}_c = \frac{\sum_{y=1}^{Y} N_{yc}}{Y}, \tag{2}$$

where $N_{yc}$ is the number of AMPs in the units that had annually been operating in the *c*-th country for 12 years, and $Y$ is the number of years ($y = 1, 2, \dots, Y$, with $Y = 12$).

The calculated values of $\overline{P}_c$ and $\overline{N}_c$ for each European country are provided in Figure 3. The length of the column indicates the average annually produced mass of HMA and WMA mixtures in millions of tonnes for 12 years in each country. The additional horizontal lines in each of the columns also present the minimum (min) and maximum (max) recorded annual amounts of the mixture produced in different countries. Respectively, from the data presented in Figure 3, it can be seen that the range between the minimum and maximum recorded annual amounts of the produced mixture in different countries is very high.

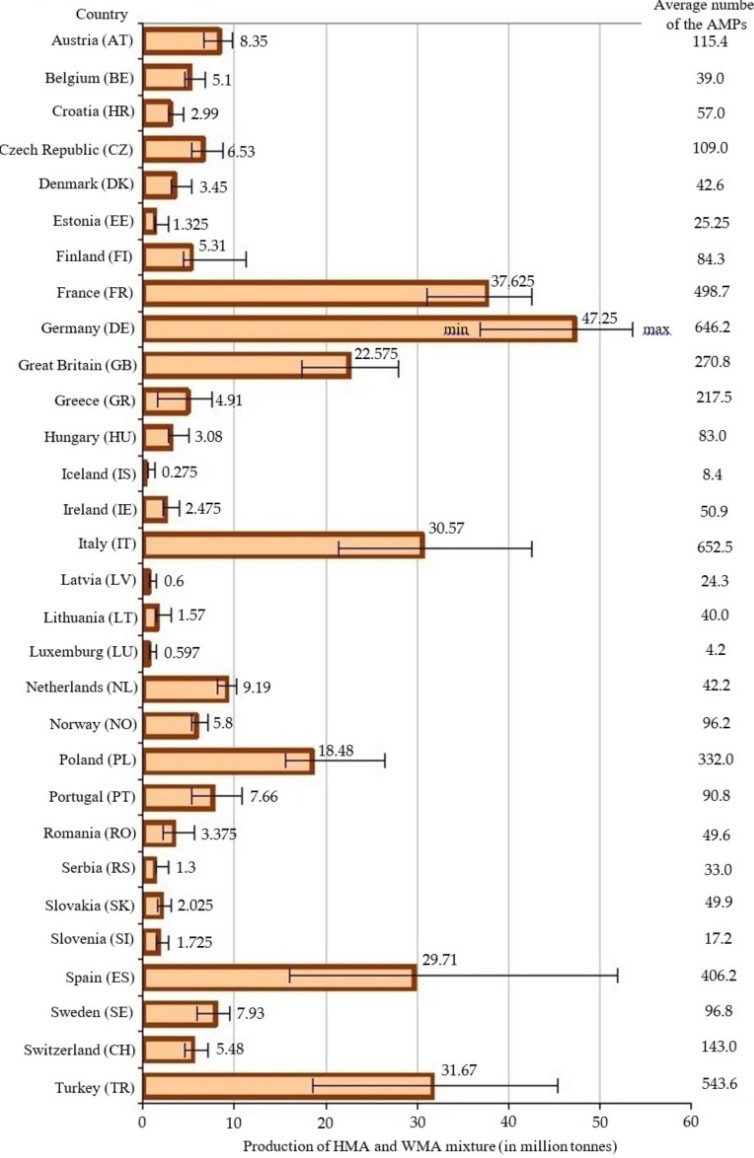

**Figure 3.** The average, maximum and minimum values of the produced quality HMA and WMA mixture in the European countries for the period of 12 years (2005 to 2016) and the average annual number of the operating AMPs.

The number of AMPs ($\overline{N}_c$) given along each column (Figure 3) was used for identifying the correlation ($\overline{P}_c$). None of the outliers were excluded during the identification of the correlation, and, as no specific non-linear correlation tendencies were noticed, a linear correlation for further analysis was selected. The positive linear correlation between $\overline{P}_c$ and $\overline{N}_c$ (Figure 4) shows that the higher the number of the operating AMPs in a country, the more HMA and WMA (coefficient of determination $R^2$ = 0.9122) is produced. The sufficiently high $R^2$ shows that the amount of asphalt mixture produced in the European countries makes approximately 91% and is subject to the number of AMPs. In comparison, only 9% of the HMA and WMA mixture is affected by other factors which are not included in the mathematical model. The points distant from the regression line show that the correlation between the factors mentioned above in some countries is different from the general trend observed in the European countries (Figure 4).

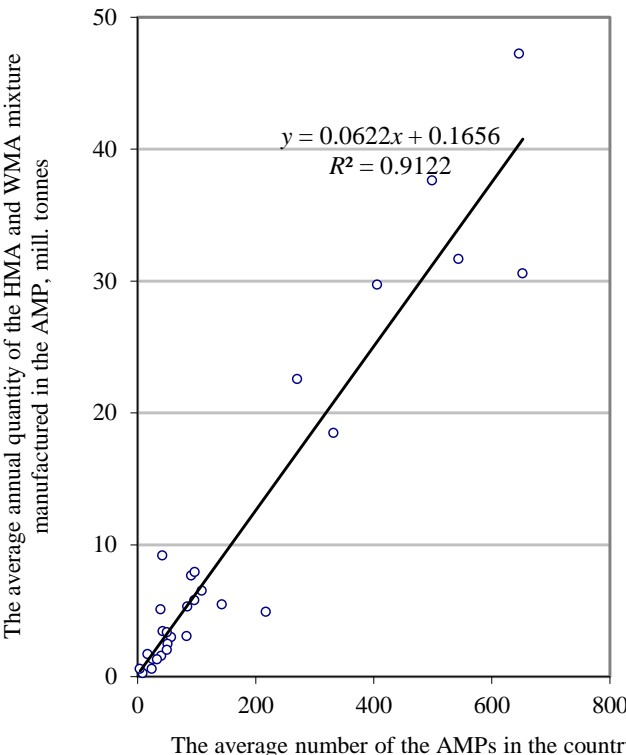

**Figure 4.** The correlation between the number of AMPs operating in 30 European counties ($\overline{N}_c$) and the amount of the asphalt mixture produced in the AMPs ($\overline{P}_c$).

The points of correlation between the average quantities of asphalt mixture and the number of AMPs installed nationwide in Belgium (BE), the Netherlands (NL), Great Britain (GB), Spain (ES), France (FR) and Germany (DE) are above the regression line (Figure 4). This indicates that the economically developed countries, as mentioned above, were among the first to install the AMPs of higher (increased) technical MC. Moreover, they were better at exploiting the technological capabilities of their AMPs in order to reduce the equipment downtime due to low demand for the asphalt mixture or bad weather conditions.

During further analysis, while taking into consideration the annual average mixture value ($\overline{P}_c$) produced by the AMPs for the period 2005–2016, all European countries have been categorized into five categories (Table 1). The threshold values of the produced asphalt values for each of the categories were selected based on the results provided in Figures 3 and 4. It must be noted that, in order to categorize the countries into different categories more precisely, the ranges of the threshold values in each of the categories are not equal. As the amount of the produced asphalt mixture increases, the range of the threshold values also increases. Correspondingly, as the amount of the produced asphalt

mixture decreases, so does the range of the threshold values. Thus, a very large amount (from 30 to 50 million tonnes) of the asphalt mixture was produced in four countries, a large quantity (from 15 to 30 million tonnes) in three countries, an average quantity (from 5 to 15 million tonnes) in nine countries, a small quantity (from 2 to 5 million tonnes) in seven countries and a minimal quantity (from 0.25 to 2 million tonnes) in seven countries (Table 1).

**Table 1.** The categorization of the countries according to the amount of asphalt mixture produced.

| The Scale of the Amount (Mass) | The Amount of the Asphalt Mixture in Millions of Tonnes and the Rank of the Quantity | The National Abbreviation and the Actual Average Amount of the Mixture Produced in all the AMPs (Millions of Tonnes/Year) |
|---|---|---|
| Very large | 30–50 (1–4) | DE (47.25); FR (37.625); TR (31.67); IT (30.57) |
| Large | 15–30 (5–7) | ES (29.71); GB (22.575); PL (18.48) |
| Average | 5–15 (8–16) | NL (9.19); AT (8.35); SE (7.93); PT (7.66); CZ (6.53); NO (5.80); CH (5.48); FI (5.31); BE (5.10) |
| Small | 2–5 (17–23) | GR (4.91); DK (3.45); RO (3.375); HU (3.08); HR (2.99); IE (2.475); SK (2.025) |
| Minimal | 0.25–2 (24–30) | SI (1.725); LT (1.57); EE (1.325); RS (1.30); LV (0.60); LU (0.597); IS (0.275) |

As for the countries with warm climates (PT, ES, IT, GR), almost no low ambient air temperature (average daily temperatures are lower than +5 °C or −3 °C) is observed, which would prevent the laying of the asphalt mixture. Therefore, the exploitation of the MC of an AMP can be higher. The use of the MC of an AMP for producing the asphalt mixture can be reduced by the national economic downturn, as many AMPs were installed during the period of economic prosperity and are currently underused (GR, LV, TS). Frequent rainfall is a meteorological factor that may also prevent the production and laying of the asphalt mixture, which results in a reduction in the level of utilisation of the MC of an AMP (GB, IS, IE). Minimising the transportation time of the asphalt mixture to the laying site can lead to a rise in the number of AMPs. This causes a drop in the MC of these facilities because the number of the devices becomes too high. The small-distance transportation of HMA mixture reduces the short-term ageing of the bitumen binder, thus making the quality of the mixture higher. Due to the fact that the resulting HMA mixture gradually loses some heat during transportation, the distance between the AMP and the laying site cannot be very long. Reducing this distance increases the required number of AMPs within the country [4]. As for the densely populated areas in the country, there is a greater need for the asphalt mixture, which leads to a larger required number of AMPs and their uneven distribution across the country. The AMPs operating in big cities are close to each other.

Greece (GR), Latvia (LV), Iceland (IS), Hungary (HU), Switzerland (CH) and Lithuania (LT) use AMPs of a lower actual MC compared to the European average. Sometimes the equipment is used irrationally because it does not produce asphalt mixture due to the downtime that lasts for a major part of the year. As a result, too many AMPs can be found in each of the above countries when the transportation time and distance of the produced asphalt mixture are sufficiently short and can be increased to acceptable limits. Competition between individual companies laying the asphalt pavement forces them to transport the asphalt mixture from their AMPs very far (as unauthorised) and for a long time despite the proximity of an AMP owned by another company. The quality of asphalt mixture transported over long distances worsens because of the cooling, density, segregation and short-term oxidation ageing of the bituminous binder.

*3.3. The Use of Asphalt Mixing Plant Capacities in Europe*

The positive linear correlation (Figure 4) shows that an increase in the number of AMPs across the country ($\overline{N}_c$) results in a rising volume of the HMA and WMA mixture

($\overline{P}_c$) produced in the plants. Regarding the introduced values, the average amount of the asphalt mixture ($Q_{c,AMP}$) annually manufactured in one AMP in each of the 30 countries, Europe and E-28 was calculated using the formula (in tonnes):

$$Q_{c,AMP} = \frac{\overline{P}_c}{\overline{N}_c}. \tag{3}$$

The results of the calculation (Figure 5) show that a single AMP annually mostly (more than 100,000 t) produces HMA and WMA mixtures in the following way: NL (217,773 t), LU (142,143), BE (130,769 t) and SI (100,291 t). From 80,000 to 100,000 tonnes is produced in PT (84,361 t), GB (83,364 t), SE (81,921 t) and DK (80,986 t). The amount of asphalt mixture close to the average of the European countries (from 55,000 t to 75,000 t) is manufactured in FR (75,446 t), ES (73,141 t), DE (73,120 t), AT (72,357 t), RO (68,044 t), FI (62,989 t), NO (60,291 t), CZ (59,908 t), TR (58,260 t) and PL (55,663 t). Less than the average of the European countries (from 40,000 t to 55,000 t) in a single AMP per one year is made in EE (52,475 t), HR (52,456 t), IE (48,625 t), IT (46,851 t) and SK (40,581 t). A lesser amount of the asphalt mixture (from 30,000 to 40,000 t) in a single AMP is manufactured in RS (39,394 t), LT (39,250 t), CH (38,322 t), HU (37,108 t) and IS (32,728 t). The least amount (less than 25,000 t) of the asphalt mixture in Europe is produced in the AMPs of LV (24,691 t) and GR (22,575 t).

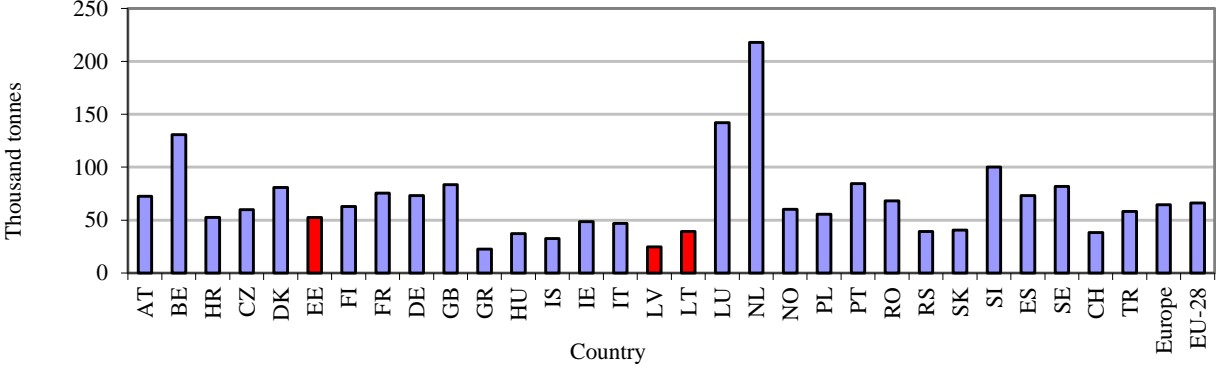

**Figure 5.** The average amount of asphalt mixture annually produced in a single AMP.

Based on the research results described above and shown in Figure 5, for a more detailed analysis, a histogram was developed to represent the frequency distribution of the average mixture production in the AMPs (Figure 6). On the vertical axis of the histogram, the widths of each interval are equal—32,533 t/year. Specifically, the histogram shows the variations in the actual annual MCs of AMPs in the European countries, which ranges from 22,575 (GR) to 217,773 t/year (NL). The difference between the maximum and minimum actual MC indicates that the use of the technical capabilities of AMPs to produce the asphalt mixture varies nearly 10-times in individual countries. The MC of new AMPs is 80–400 t/h ("Ammann") and 160–400 t/h ("Benninghoven"). The difference makes 5.0 and 2.5 times, respectively. Therefore, it is very likely that the cause for the approximately 10-times actual difference in the MCs of the AMPs is the extended downtime of the AMPs rather than significant differences in their technical mixing capacity. This statement can also be explained by the fact that an AMP usually does not produce asphalt mixtures to store them, i.e., an AMP only produces asphalt mixtures when there is a demand. That means that, if there is no demand for asphalt mixtures in specific countries, the AMPs are going to be in downtime, which can cause a significant difference in the MCs of the AMPs of the European countries.

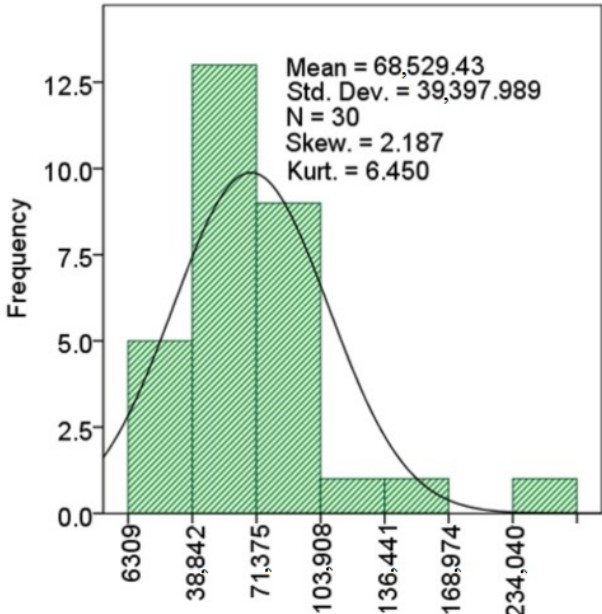

**Figure 6.** The variations in the average amount of HMA and WMA mixtures annually produced in a single AMP in Europe for the 12-year period (2005–2016).

For the 30 European countries, one AMP produces on average 68,530 tonnes of HMA and WMA mixtures a year. The standard deviation from the actual MC of an AMP is equal to approximately 39,400 t/year (Figure 6). The coefficient of variation $CoV = 57.5\%$ indicates that the distribution of the actual MC of the AMPs in individual European countries is not normal because $CoV > 33\%$. When the modules $|sk|$ and $|ku|$ of skewness and kurtosis are calculated from the averages of the actual mixing capacity ($Q_{c,AMP}$) of the AMPs in 30 countries, which are smaller than the values of $s_{sk}$ and $s_{ku}$ multiplied by 3 and 5, it can be reasonably considered that the experimental data are distributed according to the Gaussian distribution. The values $s_{sk}$ and $s_{ku}$ of the standard deviations are subject only to the sample size and are calculated according to the formulas given in [49]. While taking into consideration the sample of the 30 countries, the coefficients $3s_{sk} = 1.28$ and $5s_{ku} = 4.26$ are smaller than $|sk| = 2.189$ and $|ku| = 6.450$. A result such as this indicates that the distribution of the actual MC of the AMPs in individual countries is not normal.

The gained experience shows that the renovation of the AMPs' fleet is underway in all countries and that outdated AMPs are replaced with new technological equipment that meets modern requirements. The most significant changes in upgrading the AMPs' fleet took place in the Eastern European countries and the Baltic States, where the technically limited low-MC "Dormashina" AMPs produced in the USSR (Ukraine) had been operating. Thus, a further case study on the Baltic States is performed.

## 4. Case Study: The Development of the Asphalt Mixing Plants' Fleet in the Baltic States and the Identification of the Criteria for Making a Decision on Purchasing an Asphalt Mixing Plant

### 4.1. Historical Data

The three Baltic States (Estonia, Latvia and Lithuania) were the republics of the former Soviet Union (USSR). Therefore, the companies producing the asphalt mixture in these countries used the "Dormashina" (Kremenchuk, Ukraine) and "Teltomat" (Teltov-DDR) equipment of relatively low-MC and simple design. These plants polluted the environment, their usage of electricity and heat energy was inefficient, and the control systems were not advanced (without processors and other electronic control units). The statistical data on the operating AMPs were not annually systematically collected and analysed. Thus,

researchers showed their initiative for a certain period and counted the AMPs operating in the Baltic States. The findings were published in scientific reports, dissertations or research papers.

The economies of the Western European countries have allowed for the rapid implementation of scientific and technical progress in road building. As a result, the increasing demand for high-quality asphalt mixtures and environmental protection has forced the AMP designers and manufacturers to be ready to supply the technological equipment which would meet these requirements. As for the Eastern European countries, and the Baltic States in particular, before the restoration of independence in 1990 and the following decade, the old generation AMPs had still been operating. The dynamics of renewing the AMPs' fleet (Figure 7) shows that, over time, the ratio of the number of the new to old AMPs has gradually increased (from $t_1$) in the Baltic States, and the total number has decreased (to $t_n$) due to the purchasing of modern equipment of a significantly higher MC. This qualitative breakthrough has made it possible to significantly improve the AMPs' fleet, which has come close to the requirements raised by other European countries according to the technical indicators.

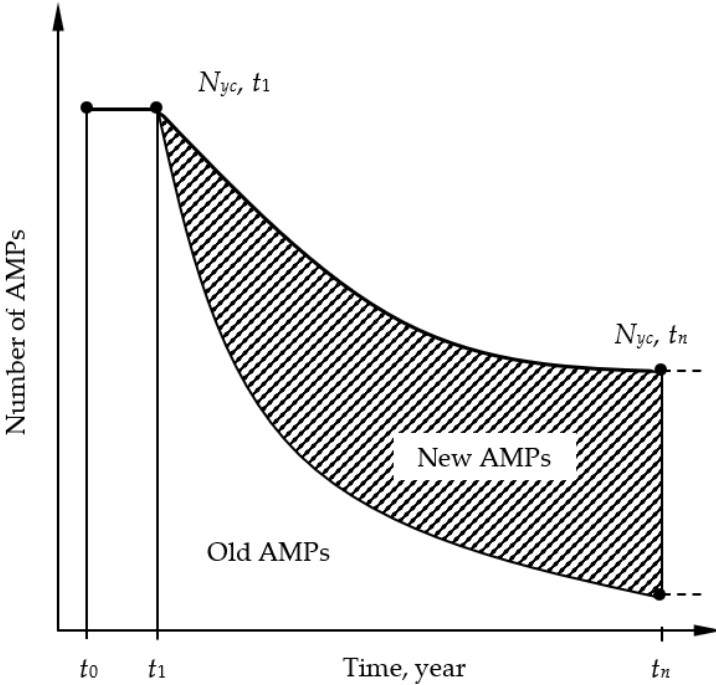

**Figure 7.** The model of the dynamics of renovating the AMPs' fleet in the Baltic States.

For example, in 1982, Lithuania counted 119 AMPs that operated in 45 asphalt production sites of the companies producing and laying the asphalt mixture. These AMPs manufactured 1.95 million tonnes of the HMA mixture. Two years later (in 1984), 48 asphalt production sites were functioning in the country and 135 stationary AMPs produced 2.284 million tonnes of the HMA mixture. Interestingly, in 1982, only five "Teltomat" AMPs (with production rates of 80 to 100 t/h) were available throughout the entire country. In contrast, the other 114 units used equipment manufactured by "Dormashina", and their production rate was from 20 to 25 t/h. During the period of renovation of the technological equipment, in 1984, there were 14 AMPs made by "Teltomat", and the rest (121 units) were manufactured by "Dormashina".

For a long time, the country did not keep records on its AMPs. For a period of 15 years, the author of this article and his colleagues collected data that allowed the determination of 99 operating AMPs in 44 asphalt production sites in Lithuania. Their activity ranged from 1 to 38 years (average 18.8 years). In 1999, 24 "Teltomat-5" AMPs operated in Lithuania. For the period 1987–1999, private business companies bought 7 new AMPs: "Ammann

Euro A240" and "160 Global H", "Marini MAP 155 E 190 L", "Benninghoven AE 150T", "Machinery AMO BS 200 / RC120", "Maschinen GmbH Teltomat-160" and "Dormashina D-645-2", the MCs of which varied from 100 to 240 t/h.

### 4.2. The Current Situation and Scenarios for Renewing the Asphalt Mixing Plants' fleet

After the restoration of independence in the Baltic States and while striving for membership in the European Union (EU), it was necessary to improve the quality of road building. Thus, both the mobile equipment fleet and the stationary AMPs producing HMA mixture were renovated. During the period of 1999–2017, the AMPs' fleet substantially improved. Almost no "Dormashina" machinery is left, as the units were replaced with new or shortly used AMPs, the number of which is listed in Table 2. The most popular 103 operating AMPs in the Baltic States represent "Teltomat" (32), "Benninghoven" (31), "Ammann" (13) and "Amomatic" (11) AMPs. Most AMPs operate in Lithuania (47) and the smallest number is observed in Estonia (23), which correlates with the area and population of the countries.

**Table 2.** The number of AMPs and their manufacturers that operated in the Baltic States in 2017.

| The Production Company | The Place of the AMP and Country | | | The Total Number of AMPs by the Company | The Production Country |
|---|---|---|---|---|---|
| | Lithuania | Latvia | Estonia | | |
| Ammann | 2 | 11 | 0 | 13 | Germany |
| Amomatic | 4 | 2 | 5 | 11 | Finland |
| Wibau | 0 | 1 | 0 | 1 | Germany |
| Benninghoven | 16 | 7 | 8 | 31 | Germany |
| Kremenchug | 5 | 0 | 0 | 5 | Ukraine |
| Marini | 1 | 3 | 0 | 4 | Italy |
| Lintec | 0 | 1 | 0 | 1 | Germany |
| Teltomat | 19 | 7 | 6 | 32 | Germany |
| Kalottikone | 0 | 0 | 3 | 3 | Finland |
| Other | 0 | 1 | 1 | 2 | China |
| Total in the countries | 47 | 33 | 23 | 103 | – |

For the last 35 years, a significant decrease in the number of asphalt production companies and AMPs has been noticed in Lithuania (Table 3).

**Table 3.** The number of APCs and AMPs in Lithuania.

| Year | 1982 | 1984 | 1999 | 2008 | 2017 |
|---|---|---|---|---|---|
| The number of APCs | 45 | 48 | 44 | 25 | 21 |
| The number of AMPs | 119 | 135 | 99 | 28 | 47 |

APC—asphalt production company.

A discussion with the experts of asphalt pavement companies has disclosed that purchasing an AMP in the Baltic States is beneficial when more than 35,000 tonnes of asphalt mixture purchased from other companies is being laid within one year.

In general, the technological equipment for producing asphalt mixture can be renovated according to one of the four cases presented below:

- A new AMP is installed beside another operating AMP (the total number of AMPs in the country increases);
- A new AMP is installed at the site of the old, dismantled AMP (the total number of AMPs in the country does not change, i.e., it remains the same);
- A new AMP is installed at a new site (the total number of AMPs in the country increases);
- An old AMP is dismantled instead of installing a new AMP, thus re-cultivating the land (the total number of AMPs in the country decreases).

One of the four cases mentioned above is selected on individual bases (sites) to produce the asphalt mixture. The experts and managers or owners of the purchasing organisation decide what kind of company (manufacturer) and what type of AMP is best to buy based on their personal experience, knowledge, recommendations, economic conditions and intuition.

AMPs are most frequently acquired and installed on the sites of already-operating plants. New and free (underdeveloped) sites are rarely used for installing AMPs. For example, in the last few years, only one asphalt production company has been provided with the possibility of installing a new AMP in Lithuania.

The Baltic States have calculated the volumes of asphalt mixture late enough. As for Lithuania, as a Member State of the EU, for the first time, 1.731 million tonnes of produced asphalt mixture were counted in 2007 [74]. This amount has been included in the EAPA database. Furthermore, based on the described case of the Baltic States, the criteria for making a decision on purchasing an asphalt mixing plant are investigated.

### 4.3. The Criteria for Purchasing Asphalt Mixing Plants

The authorised persons or owners decide to purchase a new AMP for companies producing asphalt mixtures and using them to lay the transport infrastructure objects' pavement (roads, streets, airfields). To reach this decision, every company that intends to buy an AMP has individual (own) criteria subject to specific conditions. At this point, it must be explained that the further described criteria were formulated after very in-depth discussions with the authorised persons from the companies which operate AMPs in the Baltic States. After analysing all the possible criteria for acquiring a new AMP, they have been systematised and described in linguistic form. In practice, nine ($n = 9$) cases (criteria), the significance or relative repetition rate of which has not been studied in research papers until now, are distinguished. From a theoretical point of view, a higher number of criteria could be considered. However, during the in-depth discussions with the authorised persons from the companies which operate AMPs in the Baltic States, additional criteria to the formulated nine ones were named as excessive criteria, without significant meaning. Thus, in this study, the criteria which were described as excessive are not considered.

The factors (criteria) determining the decision to purchase an AMP are as follows:

1.  An AMP operating in a road construction company has an insufficient mixing capacity, is technically outdated, energy intensive, pollutes the environment and cannot produce a mixture of all types of asphalt (a technically outdated and technologically powerless AMP).
2.  Having reliable information that construction works for a large transport infrastructure object will be carried out in the country which will require a vast amount of asphalt mixture and expecting to carry out these works (a large object of asphalt paving works is planned in the country).
3.  Due to natural disasters (fire, flood, mudflow, earthquake, rainfall, hurricane, etc.), an irreparably damaged AMP will be replaced with a new installed technological unit (a natural disaster irreparably damages an AMP).
4.  The improved economy of the country allows for a significant increase in the investment in the Road Maintenance and Development Programme, expecting success in winning public procurement contests for the asphalt paving (improved national economy, accelerating road development).
5.  The significantly improved performance indicators for a road construction company enable competitors to participate in public procurement contests for the asphalt pavement of the transport infrastructure objects (improved road construction company indicators).
6.  A neglected road transport infrastructure that does not meet the needs of society and requires high transportation expenses due to unsafe traffic, congestion, increased travel times and fuel consumption, ambient air pollution, damaging of vehicles,

underdeveloped tourism in the country and transit traffic (a transport infrastructure that does not meet the needs of society).

7. The stricter requirements to improve the quality of the produced asphalt mixture that cannot be met by the operating (outdated) AMP or can only be met partially by making special efforts and conditions during production (increased requirements for the quality of the asphalt mixture).

8. A road construction company has the opportunity (expectations) to receive support and reductions in prices from the European Structural Funds, which reduces the expenditure on purchasing a new AMP (support from the European funds is expected).

9. A business company (enterprise) expands activities by taking up the new extra business of producing (and laying) asphalt mixture (the company will additionally carry out the asphalt paving works).

### 4.4. The Weight of the Criteria

The significance of the above criteria has been assessed by a skilled expert working in a foreign capital company engaged in AMP sales and maintenance activities in the Baltic States. The expert is well aware of the criteria that have made the road construction companies of the Baltic States buy new AMPs. This allowed him, together with the authors of this work, to apply the AHP method for filling in the pairwise comparison matrix of the criteria A = $\|a_{ij}\|$, thus calculating the eigenvector (the relative significance of the criteria, i.e., the normalised weight) $\omega_i$ and checking the matrix consistency. The pairwise comparison matrix of the criteria for purchasing an AMP has been filled in by the expert and is provided in Table 4. The expert also compared all the criteria in the Saaty rating scale from 1 to 9 [75] and determined the intensity of their mutual significance from 1 to 9. The Saaty scale is a fundamental part of the AHP approach, which allows for the comparison of the criteria. The basics of the Saaty scale are described in detail in [75]. It is known [76] that a stimulus compared with itself is permanently assigned the value of 1, so the main diagonal entries of the pairwise comparison matrix are all 1. Only the integer values should be used for the comparisons. The number 3 corresponds to the verbal judgment "moderately more dominant", 5 to "strongly more dominant," 7 to "very strongly more dominant", and 9 to "extremely more dominant"; values 2, 4, 6, and 8 are used as a compromise between the previous values. If some of the criteria end up having similar mutual significance, then similar values to those criteria should be assigned. The Eigenvector $\omega_i$, showing the relative significance of the criteria was calculated according to formula (4) after normalisation:

$$\omega_i = \frac{\sqrt[n]{\prod_{j=1}^{n} a_{ij}}}{\sum_{i=1}^{n} \sqrt[n]{\prod_{j=1}^{n} a_{ij}}} \tag{4}$$

where $a_{ij}$ is the pairwise comparison matrices $\mathbf{A} = [a_{ij}]$ elements, $i, j = 1, 2, \ldots, n$; $n$ is the number of criteria; and $\omega_i$ is the weight of the $i$-th criterion.

The consistency of the pairwise comparison matrix $\mathbf{A}$ of the AMP factors has been established by calculating the consistency ratio:

$$C.R. = \frac{C.I.}{R.I.}, \tag{5}$$

where *C.I.* is the consistency index and *R.I.* is the random index.

The consistency index *C.I.* is calculated from formula (6):

$$C.I. = \frac{\lambda_{max} - n}{n - 1}, \tag{6}$$

where $\lambda_{max}$ is the largest eigenvalue, which is calculated from formula (7):

$$\lambda_{max} = \frac{1}{n} \sum_{i=1}^{n} \frac{\sum_{j=1}^{n} a_{ij}\omega_j}{\omega_i}. \qquad (7)$$

The random index *R.I.* is found in the table [75] and is subject only to the size of the pairwise comparison matrix, i.e., the number of AMP factors.

**Table 4.** The pairwise comparison matrix for identifying the significance (importance) of the criteria for purchasing a new AMP.

| | | Factor (Cause) $j = 1, 2, \dots, n$ | | | | | | | | |
|---|---|---|---|---|---|---|---|---|---|---|
| | | **A** | **B** | **C** | **D** | **E** | **F** | **G** | **H** | **I** |
| Factor (Cause) $i = 1, 2, \dots, n$ | A | 1 | 1/4 | 1/3 | 2 | 1 | 4 | 1/5 | 1/3 | 5 |
| | B | 4 | 1 | 3 | 5 | 3 | 7 | 1 | 1 | 8 |
| | C | 3 | 1/3 | 1 | 5 | 3 | 7 | 1/3 | 1 | 7 |
| | D | 1/2 | 1/5 | 1/5 | 1 | 1/2 | 3 | 1/7 | 1/5 | 4 |
| | E | 1 | 1/3 | 1/3 | 2 | 1 | 3 | 1/4 | 1/3 | 6 |
| | F | 1/4 | 1/7 | 1/7 | 1/3 | 1/3 | 1 | 1/8 | 1/7 | 2 |
| | G | 5 | 1 | 3 | 7 | 4 | 8 | 1 | 2 | 9 |
| | H | 3 | 1 | 1 | 5 | 3 | 7 | 1/2 | 1 | 7 |
| | I | 1/5 | 1/8 | 1/7 | 1/4 | 1/6 | 1/2 | 1/9 | 1/7 | 1 |

Regarding the introduced methodology, the significance of the criteria for purchasing an AMP, i.e., the subjective normalised weights ($\omega_i$), is calculated. The sequence and results of the calculations are achieved by following four steps which are provided in Table 5.

**Table 5.** The sequence (algorithm) of calculating the weight cause (criterion) Eigenvector.

| Calculation Step | A | B | C | D | E | F | G | H | I |
|---|---|---|---|---|---|---|---|---|---|
| Step one: The elements $a_{ij}$ of each line in the matrix are multiplied | $\omega_1''$ | $\omega_2''$ | $\omega_3''$ | $\omega_4''$ | $\omega_5''$ | $\omega_6''$ | $\omega_7''$ | $\omega_8''$ | $\omega_9''$ |
| | | | | | $\omega_i'' = \prod\limits_{j=1}^{9} a_{ij}$ | | | | | |
| The product of the first line | | | | $\omega_1'' = 1 \times 1/4 \times 1/3 \times 2 \times 1 \times 4 \times 1/5 \times 1/3 \times 5 = 0.2222$ | | | | | | |
| Result | 0.2222 | 1008 | 245 | 0.0034286 | 0.33333 | 0.00002025 | 60,480 | 11,025 | 0.000001181 |
| Step two: $n$-degree root is extracted from $\omega_i''$ | $\omega_1'$ | $\omega_2'$ | $\omega_3'$ | $\omega_4'$ | $\omega_5'$ | $\omega_6'$ | $\omega_7'$ | $\omega_8'$ | $\omega_9'$ |
| | | | | | $\omega_i' = \sqrt[9]{\omega_i''} = \sqrt[9]{\prod\limits_{j=1}^{9} a_{ij}}$ | | | | | |
| The root of the products of the first line | | | | | $\omega_1' = \sqrt[9]{0.22222} = 0.8461\dots$ | | | | | |
| Result | 0.8461 | 2.785 | 1.84274 | 0.5323 | 0.8851 | 0.3009 | 3.3985 | 2.8129 | 0.2195 |
| Step three: The sum of all $\omega_1'$ | | | | | $\sum\limits_{i=1}^{9} \omega_i' = \sum\limits_{i=1}^{9} \sqrt[9]{\prod\limits_{j=1}^{9} a_{ij}}$ | | | | | |
| The sum of all constituents | | | | $0.8461 + 2.7850 + 1.8427 + 0.5323 + 0.8851 + 0.3009 + 3.3985 + 2.8129 + 0.2195 = 13.623$ | | | | | | |
| Step four: Every element $\omega_i'$ is divided from the sum of all elements $\omega_i'$ of the first factor is | $\omega_1$ | $\omega_2$ | $\omega_3$ | $\omega_4$ | $\omega_5$ | $\omega_6$ | $\omega_7$ | $\omega_8$ | $\omega_9$ |
| | | | | | $\dfrac{\sqrt[9]{\prod\limits_{j=1}^{9} a_{ij}}}{\sum\limits_{i=1}^{9}\sqrt[9]{\prod\limits_{j=1}^{9} a_{ij}}} = \dfrac{\omega_i'}{\sum\limits_{i=1}^{9}\sqrt[9]{\omega_i'}}$ | | | | | |
| divided from the sum of all elements | | | | | $\omega_1 = \frac{0.8461}{13.623} = 0.0621, \dots$ | | | | | |
| The Eigenvector is obtained | 0.0621 | 0.2044 | 0.1353 | 0.0391 | 0.0650 | 0.0221 | 0.2495 | 0.2064 | 0.0161 |

**Table 5.** *Cont.*

| Calculation Step | A | B | C | D | E | F | G | H | I |
|---|---|---|---|---|---|---|---|---|---|
| The sum of the normalised weights of all factors | | | | $\sum\limits_{i=1}^{9} \omega_i = 1.0000$ | | | | | |
| The priorities (ranks) of the factors | 6 | 3 | 4 | 7 | 5 | 8 | 1 | 2 | 9 |
| Succession | | | | $G \succ H \succ B \succ C \succ E \succ A \succ D \succ F \succ I$ | | | | | |

Next, the consistency of the matrix filled in by the expert is verified (Table 5). For that purpose, the sum of the products $\sum\limits_{j=1}^{9} a_{ij}\omega_j$ is divided from the value of the calculated Eigenvector $\omega_i$ of a particular factor (criterion) (Table 5). The intermediate calculation results are presented in Table 6.

**Table 6.** The interim values for the calculation of the largest eigenvalue $\lambda_{max}$.

| The Cause for Purchasing an AMP | The Eigenvector (Relative Importance) $\omega_i$ | $\sum\limits_{j=1}^{9} a_{ij}\omega_j$ | $\dfrac{\sum\limits_{j=1}^{9} a_{ij}\omega_j}{\omega_i}$ |
|---|---|---|---|
| A | 0.0621 | 0.5891 | 9.4863 |
| B | 0.2044 | 1.9886 | 9.7290 |
| C | 0.1353 | 1.3372 | 9.8840 |
| D | 0.0391 | 0.3782 | 9.6726 |
| E | 0.0650 | 0.6126 | 9.4246 |
| F | 0.0221 | 0.2137 | 9.6697 |
| G | 0.2495 | 2.4385 | 9.7735 |
| H | 0.2064 | 1.5151 | 7.3406 |
| I | 0.0161 | 0.1623 | 10.0808 |

The arithmetic mean of the last column in Table 6 equal to the highest Eigenvalue $\lambda_{max}$ is calculated as:

$$\lambda_{max} = \frac{1}{n}\sum_{i=1}^{n}\frac{\sum_{j=1}^{n} a_{ij}\omega_j}{\omega_i}$$
$$= \frac{1}{9}(9.4863 + 9.7290 + 9.8840 + 9.6726 + 9.4246 + 9.6697 + 9.7735 + 7.3406 + 10.0808)$$
$$= 9.4478.$$

The consistency index (*C.I.*) is estimated as:

$$C.I. = \frac{\lambda_{max} - n}{n - 1} = \frac{9.4478 - 9}{9 - 1} = 0.05598.$$

The random index (*R.I.*) from the table [75] is defined or calculated according to formula (8):

$$R.I. = \frac{1.98(n - 2)}{n}, \tag{8}$$

when *n* = 9, then *R.I.* = 1.45.

The consistency ratio *C.R.* is calculated as:

$$C.R. = \frac{C.I.}{R.I.} = \frac{0.05598}{1.45} = 0.0386.$$

The calculated *C.R.* is compared with its critical (maximum allowed) value of 0.1, which is acceptable to make the pairwise comparison matrix **A** consistent. *C.R.* = 0.0386 is less than 0.1, and thus it can reasonably be considered that matrix **A** is consistent and its Eigenvector $\omega_i$ is the relative weight of the criteria.

The results of the study have shown that the road construction companies of the Baltic States purchase new AMPs mostly due to increased requirements for the quality of asphalt mixture ($\omega_G$ = 0.2495), expecting support from the European funds ($\omega_H$ = 0.2064) and when a large amount of asphalt paving is planned across the country ($\omega_B$ = 0.2044). These three key factors account for approximately two thirds (the sum of the weights is equal to 0.6603) of all nine criteria leading to the purchase of a new AMP. The rare criteria for purchasing a new AMP are observed when the company expects additional asphalt paving works ($\omega_I$ = 0.0161), the transport infrastructure does not satisfy the public needs ($\omega_F$ = 0.0221) and the improved economy of the country is accelerating the road development ($\omega_D$ = 0.0391).

## 5. Discussion and Conclusions

The decision to purchase a new asphalt mixing plant and to renew the asphalt mixing plant fleet is a complex one which requires detailed strategic planning, a SWOT analysis and the consideration of long-term effects. If the newly installed asphalt mixing plant is not going to produce asphalt mixture most of the time, the renewal of the fleet can lead to significant long-term financial losses or other undesirable consequences. In this work, it was assumed and proved that the level of the utilisation of the currently used asphalt mixing plants, i.e., the amount of the produced asphalt mixture in the currently used asphalt mixing plants, is a proper indicator to define the actual need for new asphalt mixing plants in the country. During the analysis it was found that the annual average amount of the asphalt mixture produced across 30 European countries during the period of the last 12 years (2005 to 2016) ranged from 0.3 million (Iceland) to 47.25 million tonnes (Germany). The estimated amount of asphalt mixture produced per asphalt mixing plant in each of the 30 European countries ranges from 22,575 (Greece) to 217,773 tonnes (the Netherlands). Almost a 10-times difference in the average amount of asphalt mixture produced in a single AMP, while taking into consideration different European countries, can be seen. Based on the performed research, it is very likely that the reason for an almost 10-times difference between the asphalt mixtures produced in individual countries is due to the varying mixing capacities of their technological facilities. It was determined that the average mixing capacity for all the asphalt mixing plants in the European countries is 68,530 t/year, the standard deviation of the mixing capacity is 39,400 t/year, and the data distribution is not normal. Furthermore, the analysis showed that the lowest amounts of asphalt mixture produced in Europe are those that come out of the asphalt mixing plants of Lithuania (39,250 t), Switzerland (38,322 t), Hungary (37,108 t), Iceland (32,728 t), Latvia (24,691 t) and Greece (22,575 t). As a result, it can be stated that too many asphalt mixing plants can be found in each of these countries. Based on these results, it can be stated that the ongoing renovation of the asphalt mixing plants' fleet in Europe needs to be reconsidered. The need for clearly defined criteria to serve as guidelines for the decision-making process of an asphalt mixing plant's fleet renewal strategy provides an important research problem. This works contributes to the solving of this issue. The main contribution of this work is the proposal of a novel nine criteria system applying to the decision-making process for the purchase of an asphalt mixing plant. The significance of these criteria was determined after the presentation of a case study.

Based on this research, the following conclusions are formulated:

1. The research has shown that the differences between the minimum and maximum annual amounts of asphalt mixture produced and the differences between the average annual numbers of the operating asphalt mixing plants in different European countries are very high. Based on the significant difference, asphalt mixture production across the 30 European countries was categorized into five categories: (1) very large quantities (from 30 to 50 million tonnes, four countries in Europe); (2) large quantities (from 15 to 30 million tonnes, three countries); (3) average quantities (from 5 to 15 million tonnes, nine countries); (4) small quantities (from 2 to 5 million tonnes, seven countries); and (5) minimal quantities (from 0.25 to 2 million tonnes, seven

countries). The average annual minimum number of asphalt mixing plants in operation in these countries was 4.2 (Luxembourg) and the highest average number was 652.2 (Italy).

2.  It was determined that the correlation between the number of asphalt mixing plants and the production of asphalt mixture in European countries is a positive linear correlation with the coefficient of determination $R^2 = 0.9122$. Such a sufficiently high $R^2$ proves that the asphalt mixture produced in European countries is subject to the number of asphalt mixing plants. Only 9% of the asphalt mixture produced in European countries is related to factors other than the number of asphalt mixing plants.

3.  The nine most important criteria that have an impact on the decision to purchase a new asphalt mixing plant were determined and described. During the analysis of the Baltic States, it was determined that, most commonly, the Baltic States purchase new asphalt mixing plants while taking into consideration the following criteria: (1) increased requirements for the quality of asphalt mixture (weight of the criteria—0.2495); (2) when expecting support from the European funds (weight of the criteria—0.2064); and (3) when a large object of asphalt paving is planned across the country (weight of the criteria—0.2044). These three key criteria account for approximately 66% of all nine criteria for renewing the asphalt mixing plants' fleet. The other six reasons account for 34% of all the purchased asphalt mixing plants.

In general, the results of this study can serve as guidelines for stakeholders while seeking to precisely plan the AMPs' fleet renewal strategy. However, there are also some limitations. These, which will be considered in future research, are as follows:

1.  While developing the novel nine criteria system, the actual technical properties of the asphalt mixing plants (e.g., the design of dryer drums, the efficiency of dust collection, the type of mixers used, the level of noise, etc.) were not considered. This limitation can be explained by the fact that the technical properties of an asphalt mixing plant have no direct influence on the decision-making process, while taking into consideration the previously described possible cases of the asphalt mixing plants' fleet renewal. However, the technical properties of an asphalt mixing plant can significantly affect the fleet renewal strategy from two different points of view: the expenses/price and the efficiency of the asphalt mixing plant's performance. For this reason, the next step for future research by the authors is a detailed analysis of the technical parameters of the asphalt mixing plants and their influence on the fleet renewal strategy. As an extension of this work, the authors are planning to develop an additional twenty criteria system which will define the significance of the technical properties of the asphalt mixing plants for the fleet renewal strategy of these plants.

2.  The cost-benefit analysis was not performed or considered as a separate criterion in the developed nine criteria system. The need of the cost-benefit analysis could be divided into two separate groups: (1) the consideration of the long-term effects, i.e., if the renewal of the asphalt mixing plant's fleet is going to cause financial losses or gains; and (2) the price of the asphalt mixing plant, the possibility to get a discount, etc. In this work, while developing the nine criteria system, it was assumed that the renewal would cause financial losses if the number of the asphalt mixing plants in the country was too high and the new plant was going to be in downtime for the major part of the year. However, during the decision-making process, an actual cost-benefit analysis could serve as an additional support for the proposed novel nine criteria system. Regarding the future research, the authors are planning to consider the price of the asphalt mixing plant as a criterion when describing the technical parameters of the plant.

3.  In this work, the environmental consequences of asphalt mixture production due to greenhouse gas emissions were considered as an integral part of the developed expenses' model and the nine criteria system (in the A and F criteria) but not as a separate factor. One of the ways to better include the consideration of environmental consequences in the decision-making process is through the cost–benefit analysis.

Another way through which the environmental consequences could be entered into the decision-making process, which will be used by the authors in their future research of the authors, would be to consider the generated greenhouse gas emissions as a criterion which describes the performance of the asphalt mixing plant.

**Author Contributions:** Conceptualisation, H.S.; methodology, H.S.; investigation, H.S. and V.S.; writing—original draft preparation, H.S. and V.S.; writing—review and editing, P.S. All authors have read and agreed to the published version of the manuscript.

**Funding:** This research received no external funding.

**Institutional Review Board Statement:** Not applicable.

**Informed Consent Statement:** Not applicable.

**Data Availability Statement:** Datasets analyzed are available from the corresponding author.

**Conflicts of Interest:** The authors declare that there is no conflict of interest regarding the publication of this article.

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
