# Peer review of "The Correlation between the Number of Asphalt Mixing Plants and the Production of Asphalt Mixtures in European Countries and the Baltic States"

_applsci, doi:10.3390/app11209375_

Round 1

Reviewer 1 Report

The article titled "The Correlation between the Number of the Asphalt Mixing Plants and the Production of the Asphalt Mixture in European Countries and the Baltic States" focuses on the potential correlation between the number of asphalt plants in European countries and asphalt production.

Although the authors have tried to find information in many European countries and the results obtained are of importance in this area, in my opinion, the article is very flat on this topic.

English language and style are fine/minor spell check required.

The manuscript topics fit enough to the journal scope.

A large number of previous researches by the authors and others have been discussed and those results have been compared to the results of the current research.

The conclusions section should be strengthened. The authors should clearly highlight the limitations of this study and how they will be addressed in future research.

Author Response

Dear Reviewer,

We sincerely thank You for Your comments and remarks. Respectively, the provided comments were taken into consideration and the manuscript was improved according to those comments.

Other improvements of the manuscript were also performed. In the revised version of the manuscript, the literature review was improved while using additional 11 references.

More detailed answers to the comments:

COMMENT 1: Although the authors have tried to find information in many European countries and the results obtained are of importance in this area, in my opinion, the article is very flat on this topic.

ANSWER 1: Thank You for Your comment. In order to more clearly explain the logic behind our research and the contribution of the manuscript to the research field, in the introduction of the manuscript, a research framework diagram was added (Figure 1) and described. Also, the contribution of the manuscript to the research field was explained more clearly in the conclusions. However, one of the reasons why this research work may look flat is that this manuscript is a first part of the continuous research, i.e., the entire research consists of two directly related parts:

  1. The first part (this manuscript) analyses the more global situation, related to the AMPs’ fleet renewal strategy in Europe countries and the Baltic States. In this manuscript, nine most important criteria, which have an impact on the decision to purchase a new asphalt mixing plant, were determined and described.
  2. The second part (a second manuscript, which is currently being reviewed in another research journal) analyses the problem of the AMPs’ fleet renewal strategy from a different point of view. In general, the second manuscript presents a detailed analysis of the technical parameters of the asphalt mixing plants and their influence on the fleet renewal strategy.

Thus, both research works make a detailed research. Respectively, in the conclusions of the manuscript (in the part regarding the limitations of the study), the second manuscript is mentioned as a future research.

COMMENT 2: English language and style are fine/minor spell check required.

ANSWER 2: Thank You for Your comment. The manuscript was revised by a native English-speaking person. Thus, the grammar was improved; the used terminology was improved; long sentences were shortened; the punctuation, spelling and overall style were also revised.

COMMENT 3: The conclusions section should be strengthened. The authors should clearly highlight the limitations of this study and how they will be addressed in future research.

ANSWER 3: Thank You for Your comment. The limitations of the study were explained in a detailed way. In general, the entire section of the conclusions was re-written. The name of the section was changed to “Discussion and Conclusions”. The mind-flow of the conclusions was entirely changed. As it is now, in the revised version of the manuscript, the conclusions consist of three main parts: 1. Discussion; 2. Main conclusions (main 3 generalized conclusions); 3. Limitations of the study (main 3 generalized limitations of the study) and the future research directions. These limitations of the study were named and discussed in the conclusions: 1. The technical properties of the asphalt mixing plants were not considered; 2. The cost-benefit analysis was not performed; 3. The environmental consequences were not considered as a separate factor during the analysis.

Reviewer 2 Report

  • the topic of your manuscript is not clear. It is impossible to deduct what this paper is about.
  • I can`t really see the usefulness of this paper.
  • Regarding previous point, I can`t see how this paper can be useful because you don`t have conclusions. Please provide at least three general conclusions.
  • You should work on the language. It is hard to read.

Author Response

Dear Reviewer,

We sincerely thank You for Your comments and remarks. Respectively, the provided comments were taken into consideration and the manuscript was improved according to those comments.

Other improvements of the manuscript were also performed. In the revised version of the manuscript, the literature review was improved while using additional 11 references. Also, in the Introduction section, a research framework diagram was added (Figure 1) and described.

More detailed answers to the comments:

COMMENT 1: The topic of your manuscript is not clear. It is impossible to deduct what this paper is about. I can`t really see the usefulness of this paper.

ANSWER 1: Thank You for Your comment. Regarding the topic of the manuscript: In general, the need of the clearly defined criteria which would serve as guidelines for the decision-making process of the asphalt mixing plant’s fleet renewal strategy is an important research problem. This works contributes to solving this issue. Respectively, the main contribution of this work – it proposes and clarifies a novel nine criteria system that affects the decision-making to purchase an asphalt mixing plant. The results of this study can serve as guidelines for the stakeholders while seeking to precisely plan the AMPs’ fleet renewal strategy.

Also, in order to more clearly explain the logic behind our research and the contribution of the manuscript to the research field, in the introduction of the manuscript, a research framework diagram was added (Figure 1) and described. Also, the contribution of the manuscript to the research field was explained more clearly in the conclusions.

COMMENT 2: I can`t see how this paper can be useful because you don`t have conclusions. Please provide at least three general conclusions.

ANSWER 2: Thank You for Your comment. The entire section of the conclusions was re-written. The name of the section was changed to “Discussion and Conclusions”. The mind-flow of the conclusions was entirely changed. As it is now, in the revised version of the manuscript, the conclusions consist of three main parts: 1. Discussion; 2. Main conclusions (main 3 generalized conclusions, as suggested); 3. Limitations of the study (main 3 generalized limitations of the study) and the future research directions. These limitations of the study were named and discussed in the conclusions: 1. The technical properties of the asphalt mixing plants were not considered; 2. The cost-benefit analysis was not performed; 3. The environmental consequences were not considered as a separate factor during the analysis.

COMMENT 3: You should work on the language. It is hard to read.

ANSWER 3: Thank You for Your comment. Respectively, the manuscript was revised by a native English-speaking person. Thus, the grammar was improved; the used terminology was improved; long sentences were shortened; the punctuation, spelling and overall style were also revised.

Reviewer 3 Report

The paper has been well improved respect to te previous version, good job 

Author Response

Dear Reviewer,

We sincerely thank You for Your positive comment.

In any case, various improvements were still made in the manuscript: the literature review was extended with additional 11 references; a research framework diagram was added (Figure 1) and described; the conclusions were re-written; the language of the manuscript was also revised.

Round 2

Reviewer 2 Report

Thank you for addressing my comments.